# Reversible GNS for Dissipative Fluids with Consistent Bidirectional Dynamics

## Abstract

Simulating physically plausible trajectories toward user-defined goals is a fundamental yet challenging task in fluid dynamics. While particle-based simulators can efficiently reproduce forward dynamics, inverse inference remains difficult, especially in dissipative systems where dynamics are irreversible and optimization-based solvers are slow, unstable, and often fail to converge. In this work, we introduce the Reversible Graph Network Simulator (R-GNS), a unified framework that enforces bidirectional consistency within a single graph architecture. Unlike prior neural simulators that approximate inverse dynamics by fitting backward data, R-GNS does not attempt to reverse the underlying physics. Instead, we propose a mathematically invertible design based on residual reversible message passing with shared parameters, coupling forward dynamics with inverse inference to deliver accurate predictions and efficient recovery of plausible initial states. Experiments on three dissipative benchmarks (Water-3D, WaterRamps, and WaterDrop) show that R-GNS achieves higher accuracy and consistency with only one quarter of the parameters, and performs inverse inference more than 100× faster than optimization-based baselines. For forward simulation, R-GNS matches the speed of strong GNS baselines, while in goal-conditioned tasks it eliminates iterative optimization and achieves orders-of-magnitude speedups. On goal-conditioned tasks, R-GNS further demonstrates its ability to complex target shapes (e.g., characters "L" and "N") through vivid, physically consistent trajectories. To our knowledge, this is the first reversible framework that unifies forward and inverse simulation for dissipative fluid systems.

## 1 Introduction

Inverse problems are central to many fields, enabling the recovery of hidden causes from observable outcomes. For instance, estimating physical parameters from simulation results in physics or inferring joint motions from actions in robotics (Li et al., 2023a; Dinh et al., 2016; Kingma & Dhariwal, 2018; Ghosh et al., 2022; Zhang et al., 2024). Yet their application to complex systems is challenging: reversible networks impose strict constraints, making them difficult to extend to high-dimensional systems. (Ardizzone et al., 2018; Gomez et al., 2017; Behrmann et al., 2019), In fluid simulation, while forward models can efficiently reproduce flows, waves, and turbulence, it remains hard to infer an initial state that plausibly evolves into a desired outcome, such as smoke rising into a dragon-shaped cloud or waves forming characters on a shoreline.

Such goal-conditioned problems in fluid dynamics are particularly challenging. Most natural fluids are dissipative systems governed by irreversible dynamics, which makes inverse inference inherently ill-posed. Moreover, fluids are often modeled as particle systems (Sanchez-Gonzalez et al., 2020; Prantl et al., 2022), where each particle carries its own physical attributes and interacts with a large number of neighbors, leading to extremely high system complexity. Existing approaches struggle to directly recover plausible past states of dissipative fluids (Beck et al., 2024; Lefever, 2018; Levashova et al., 2021). Optimization-based methods rely on differentiable simulators (Hu et al., 2019; Holl et al., 2020; Schoenholz & Cubuk, 2020; Li et al., 2023b) to iteratively refine initial conditions, but the process is computationally expensive, unstable, and becomes increasingly inaccurate or fails to converge as particle counts and inverse steps grow. Data-driven alternatives, such as feed-forward neural simulators (Sanchez-Gonzalez et al., 2020; Prantl et al., 2022; Shao et al., 2022), can generate

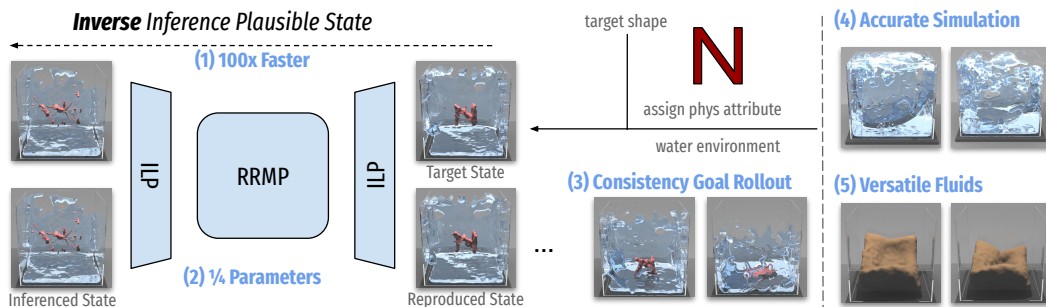

Figure 1: The pipeline illustrates unified bidirectional simulation: dashed arrows denote inverse inference and solid arrows denote forward simulation. Key advantages are highlighted along the pipeline: (1) **100× Faster**: inverse inference is orders of magnitude faster than optimization-based solvers; (2) **¼ Parameters**: a unified reversible network integrates forward and inverse reasoning with only one quarter of the parameters compared to two seperately trained GNSs; (3) **Consistent Goal Rollout**: goal-conditioned rollouts reproduce target shapes with high consistency; (4) **Accurate Simulation**: R-GNS achieves high accuracy on dissipative fluids dataset with up to 13k particles; (5) **Versatile Fluids Modeling**: Beyond water, R-GNS effectively handles other dissipative fluids like sand, highlighting its robustness across material types.

inverse solutions more efficiently, yet directly fitting inverse data often causes the model to deviate from real physical principles and yields inconsistencies with forward simulations.

To address these challenges, we introduce Reversible Graph Network Simulator (R-GNS), a compact architecture that unifies forward and inverse simulation within a single framework. Importantly, R-GNS does not assume that dissipative fluids are physically reversible; rather, it uses a mathematically reversible design to enforce forward–inverse consistency, recovering one plausible initial trajectory consistent with forward dynamics even when the underlying system is irreversible. By leveraging a reversible residual message-passing design, R-GNS propagates particle interactions in both directions while maintaining strict consistency. With shared parameters and a bidirectional training scheme, the model efficiently learns forward dynamics and simultaneously exploits this knowledge to guide inverse inference, ensuring physical plausibility and coherence between the two tasks.

We evaluate R-GNS on three dissipative fluid benchmarks: Water-3D, WaterRamps, and WaterDrop (Sanchez-Gonzalez et al., 2020). And demonstrate state-of-the-art accuracy, efficiency, and consistency in both forward and inverse simulation. On a goal-conditioned character-shaping task, R-GNS successfully generates physically plausible trajectories to match target shapes.

In summary, our contributions are: (1) a unified reversible framework that achieves higher accuracy with one quarter of the parameters, and improves forward–inverse consistency via shared parameters and bidirectional training; (2) a feed-forward solution for inverse simulation in dissipative systems, which maintains forward speed comparable to strong GNS baselines in standard rollouts, while delivering orders-of-magnitude speedups on inverse tasks by eliminating iterative optimization, and (3) a residual reversible message-passing network that guarantees invertibility, enables faithful bidirectional propagation, and better captures underlying physical laws.

## 2 RELATED WORK

**Particle-based Fluid Simulation.** Particle-based modeling has long been a powerful paradigm for simulating complex physical systems, from molecular dynamics to large-scale fluids, where each particle encapsulates its own state and interacts with many others. Traditional physics-based solvers, such as SPH and MPM (Monaghan, 2005; Sulsky et al., 1995), can deliver highly accurate forward dynamics, but require careful manual specification of fluid parameters and simulation settings, which limits their accessibility and flexibility.Recent neural simulators (Sanchez-Gonzalez et al., 2020; Shao et al., 2022; Ummenhofer et al., 2019; Prantl et al., 2022; Toshev et al., 2024; Satorras et al., 2021; Toshev et al., 2023; 2024), such as GNS and DMCF, alleviate this burden by implicitly learning particle interactions from data, enabling efficient approximations of dynamics; however, they often

struggle to maintain strict physical consistency. Moreover, the inherent complexity of particle systems varies significantly across scenarios, motivating our choice of datasets that span different levels of difficulty and setting the stage for methods that can adapt across such scales.

**Optimization-based Inverse Solvers.** Optimization-based inverse solvers built upon differentiable simulation frameworks (e.g., DiffTaichi, PhiFlow, Jax-MD) (Hu et al., 2019; Holl et al., 2020; Schoenholz & Cubuk, 2020; Li et al., 2023b) typically recover initial states through iterative optimization algorithms such as Adam or L-BFGS (Kingma, 2014; Nocedal, 1980; Liu & Nocedal, 1989). These approaches refine candidate trajectories step by step via gradient-based updates, but the process is computationally expensive and sensitive to instability. The challenge becomes more pronounced in particle-based systems, where the extremely high dimensionality exacerbates the difficulty of optimization. In dissipative fluids, where multiple plausible solutions exist and the dynamics are inherently irreversible, optimization often fails to converge, further limiting its practical applicability and underscoring the need for alternatives beyond purely iterative solvers.

**Invertible / Reversible Neural Network.** Invertible and reversible neural networks (Ardizzone et al., 2018; Gomez et al., 2017; Behrmann et al., 2019) provide a principled way to construct architectures where inputs can be exactly recovered from outputs. Such designs have been widely adopted in areas such as density estimation, robotics, and physical parameter estimation (Li et al., 2023a; Dinh et al., 2016; Kingma & Dhariwal, 2018). However, these models often rely on strong symmetry assumptions or restrictive architectural designs, which limit their flexibility in practice. The high dimensionality and large particle counts of fluid systems make reversible simulation particularly challenging, motivating the design of frameworks that ensure exact invertibility while scaling to complex dynamics.

## 3 METHODOLOGY

### 3.1 PROBLEM FORMULATION

Fluids can be represented as a particle-based system consisting of $N$ particles, the state at time step $t$ is denoted by $\chi^t = \{x_i^t\}_{i=1}^N, x_i^t = [p_i^t, m_i]$, where $p_i^t$ represents the position of particle $i$ at time $t$, and $m_i$ denotes its static attributes (e.g., material type, mass). We introduce a bidirectional simulator consisting of a forward operator $\phi$ and its inverse counterpart $\psi$. (1) **Forward simulation**: Given an initial state $\chi^0$, the simulator produces a trajectory by recursively applying $\phi$: $\tilde{\chi}^{t+1} = \phi(\tilde{\chi}^t), t = 0, ..., K - 1$. This process corresponds to standard rollouts in particle-based fluid simulation and is consistent with existing neural simulators. (2) **Inverse inference**: Unlike optimization-based methods that solve for an initial condition through iterative refinement, we instead infer it directly by applying the inverse operator $\psi$: $\tilde{\chi}^t = \phi(\tilde{\chi}^{t+1}), t = K - 1, ..., 0$. This formulation enables efficient backward reasoning and provides trajectories that are inherently tied to the forward dynamics. In both directions, we denote the resulting trajectory compactly as a "rollout", either forward ($\tilde{\chi}^{0:K}$) or inverse ($\tilde{\chi}^{K:0}$), depending on the operator applied.

In this work, we aim to learn a bidirectional simulator $\mathcal{F}_\theta$ that unifies the forward operator $\phi$ and the inverse operator $\psi$, trained from particle trajectories and capable of generalizing to unseen environments and initializations. The model is expected to satisfy three requirements: (1) Accurate forward dynamics: produce faithful, long-horizon rollouts. (2) Consistent inverse inference: recover plausible initial states aligned with forward trajectories. and (3) Robust generalization: adapt to novel particle configurations and boundary conditions without retraining.

This formulation underscores the central challenge: while existing neural simulators achieve efficiency, they often sacrifice forward–inverse consistency; meanwhile, optimization-based solvers remain unstable and computationally costly. Our objective is therefore to design a unified framework that performs both forward and inverse simulation within a single reversible architecture.

### 3.2 PRELIMINARY

**Graph Network Simulator (GNS)**   GNS (Sanchez-Gonzalez et al., 2020) is a neural particle simulator where particles are represented as nodes and their interactions as edges. Its architecture consists of an encoder MLP, a multi-layer message-passing (Gilmer et al., 2017; Scarselli et al., 2008) core, and a decoder MLP. The message-passing update is given by:

$$n_i^{l+1} = n_i^l + f^n(n_i^l, \sum_{j \in \mathcal{N}_i} f^e(n_i^l, n_j^l, e_{ij})),$$
(1)

where $n_i^l$ denotes the feature of particle $i$ at layer $l$, $e_{ij}$ are edge features, and $f^n, f^e$ are MLP layers.

**Reversible Networks (RevNet)** The vanilla RevNet achieves bijective transformations between two partitions of an input vector through dimensional splitting and dual mappings (Gomez et al., 2017). Specifically:

$$\begin{cases} y_1 = x_1 + f(x_2) \\ y_2 = x_2 + g(y_1) \end{cases}, \qquad \begin{cases} x_2 = y_2 - g(y_1) \\ x_1 = y_1 - f(x_2) \end{cases}.$$
(2)

This design ensures exact invertibility at each layer and forms the basis of reversible architectures.

These preliminaries lay the foundation for our Reversible Graph Network Simulator (R-GNS), introduced in the next section.

### 3.3 REVERSIBLE GRAPH NETWORK SIMULATOR

We propose the Reversible Graph Network Simulator (R-GNS), a unified simulator $\mathcal{F}_\theta$ that performs both forward dynamics $\phi$ and inverse inference $\psi$ within a single reversible architecture. Compared to separate neural simulators R-GNS achieves higher accuracy and stronger forward–inverse consistency with only one quarter of the parameters, while offering orders-of-magnitude faster inverse inference than optimization-based approaches. The design is particularly suited for dissipative systems, where irreversibility and multiple-solution ambiguity make inverse inference ill-posed.

As illustrated in Fig. 2, our framework consists of three tightly coupled components: (1) Semi-Symmetric Input–Output Design. In panel (a), the input graph encodes velocity history, current positions, and static attributes, while the output is masked to retain only the predicted velocity. This design preserves structural symmetry required for reversibility while reducing the difficulty of mapping by updating only dynamic quantities. (2) Invertible Linear Projection (ILP) Encoder/Decoder. Shown at the transition between the physics and latent spaces in panel (b), the projection consists of a linear mapping and its pseudo-inverse, enabling an exactly reversible transformation between physical states and latent features. (3) Residual Reversible Message Passing (RRMP). At the center of panel (b), RRMP propagates particle interactions through a mathematically invertible residual message-passing network with shared parameters. This backbone guarantees layerwise reversibility, enforces bidirectional consistency at scale, and reduces parameter redundancy.

#### 3.3.1 SEMI-SYMMETRIC INPUT–OUTPUT DESIGN

A central difficulty in combining reversible networks with physical simulators is reconciling two conflicting requirements: reversible architectures demand symmetric input–output structures, while physical dynamics are inherently asymmetric. This conflict is particularly evident in particle systems, where particles carry static attributes as well as dynamic quantities.

To address this, we propose a **semi-symmetric input-output design**: both input and output are represented as structurally symmetric graph data, satisfying the reversibility requirement, but only node dynamic quantity are predicted while other quantities are masked. In practice, the encoder consumes the complete graph state, while the decoder outputs only the motion-related node updates needed to advance the system.

As shown in Fig. 2(a), at time step $t$, forward simulation and inverse inference share the same physical attributes and particle positions, while dynamic quantities (velocities) are treated symmetrically. We encode relative particle positions as edge features and dynamic quantities as node features. During updates, edge features remain fixed as conditions, while node features are updated to capture dynamics. This design preserves structural symmetry and exact reversibility, while remaining consistent with the intrinsic update rules of physical dynamics.

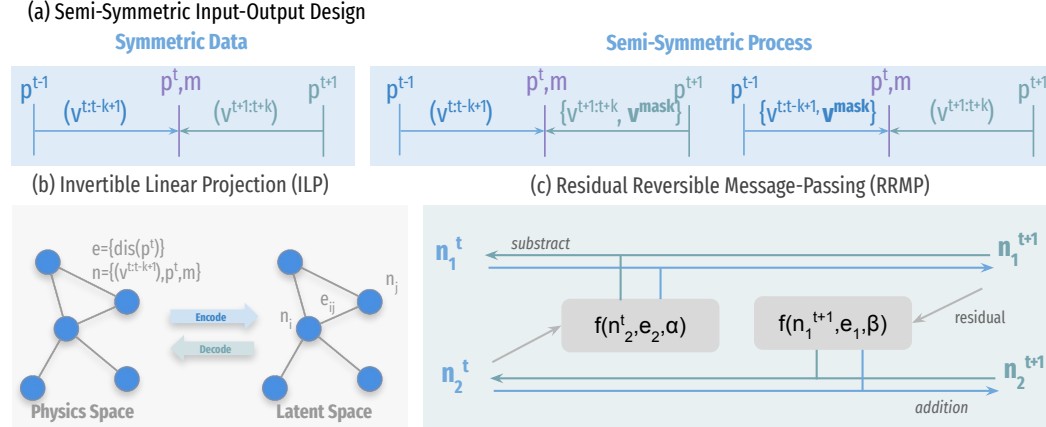

Figure 2: Overview of the R-GNS framework. (a) Semi-Symmetric Input–Output Design: Static quantities $(p, m)$ remain fixed, while dynamic quantities are symmetric in time. The model consumes full velocity history $(v^{t:t-k+1})$ but predicts only $v^{t+1}$, with other outputs masked. Positions are updated to $p^{t+1}$ using $p^t$ and $v^{t+1}$. The inverse process mirrors this update, preserving symmetry and reversibility. (b) Invertible Linear Projection: establishes a mathematically reversible mapping between physical and latent spaces, preserving all information in structured particle states. (c) Residual Reversible Message Passing: splits node dynamics into complementary halves and updates them via residual connections conditioned on fixed edges, enabling exact bidirectional propagation of latent states.

### 3.3.2  INVERTIBLE LINEAR PROJECTION EN/DECODER

Building on the semi-symmetric design, we introduce an Invertible Linear Projection (ILP) encoder–decoder to map between the physical state space and the latent feature space. Unlike learned nonlinear approximations, ILP is defined as a linear transformation paired with its pseudo-inverse:

$$\boldsymbol{n} = W\boldsymbol{\chi} + \boldsymbol{B} \quad \text{(encode)}, \qquad \boldsymbol{\chi} = W^\dagger(\boldsymbol{n} - \boldsymbol{B}) \quad \text{(decode)}, \tag{3}$$

where $\boldsymbol{\chi}$ and $\boldsymbol{n}$ denote physical and latent node features respectively, $W$ and $\boldsymbol{B}$ are learnable parameters, and $W^\dagger$ is the pseudo-inverse of $W$.

This construction guarantees exact reversibility up to numerical precision, with empirical reconstruction error bounded by $||\chi - \text{dec}(\text{enc}(\chi))||_2 \approx 10^{-7}$. As shown in Fig 2(b), ILP provides a stable bridge between physical states and latent features, allowing dimension changes without information loss. ILP is deliberately chosen over nonlinear encoders to avoid approximation drift, improving training stability and ensuring that reversible propagation is mathematically exact.

### 3.3.3  RESIDUAL REVERSIBLE MESSAGE PASSING

Building on the semi-symmetric design and ILP, the latent space consists of symmetric node features $\{n_i^t\}_{i=1}^N$, $\{n_i^{t+1}\}_{i=1}^N$ and fixed edge features $\{e_{ij}\}$. Our objective is to define a conditional reversible process:

$$\text{forward}(\{n_i^t\}, \{e_{ij}\}) \mapsto \{n_i^{t+1}\}, \quad \text{inverse}(\{n_i^{t+1}\}, \{e_{ij}\}) \mapsto \{n_i^t\}, \tag{4}$$

where $\{n\}, \{e\}$ denote the entire system, while $n_i, e_{ij}$ denote individual node and edge features. note that edge features remain fixed since particle positions are identical across forward and inverse steps (see Sec. 3.3.1).

While standard RevNets achieve invertibility through dimensional splitting, they operate on flat vectors and cannot directly handle graph-structured states with conditional edge interactions. Moreover, the latent representation is a high-dimensional matrix $(N \times d)$, making direct reversible mapping infeasible.

To overcome these challenges, we introduce Residual Reversible Message Passing (RRMP) 2(c), which extends RevNet-style updates to graph-structured data by embedding message-passing opera-

tions inside reversible residual blocks. Following the standard RevNet formulation, node features are split into two partitions $n_i^l \rightarrow (n_{i,1}^l, n_{i,2}^l)$, with two coupled functions $f$ and $g$ applied in an alternating manner. Specific, the forward update is:

$$\begin{cases} n_{i,1}^{l+1} = n_{i,1}^l + f^n(n_{i,2}^l, \sum_{j \in \mathcal{N}_i} f^e(n_{i,2}^l, n_{j,2}^l, e_{ij,2})) \\ n_{i,2}^{l+1} = n_{i,2}^l + g^n(n_{i,1}^{l+1}, \sum_{j \in \mathcal{N}_i} g^e(n_{i,1}^{l+1}, n_{j,1}^{l+1}, e_{ij,1})) \end{cases}, \tag{5}$$

and the corresponding inverse update is:

$$\begin{cases} n_{i,2}^l = n_{i,2}^{l+1} - g^n(n_{i,1}^{l+1}, \sum_{j \in \mathcal{N}_i} g^e(n_{i,1}^{l+1}, n_{j,1}^{l+1}, e_{ij,1})) \\ n_{i,1}^l = n_{i,1}^{l+1} - f^n(n_{i,2}^l, \sum_{j \in \mathcal{N}_i} f^e(n_{i,2}^l, n_{j,2}^l, e_{ij,2})) \end{cases}. \tag{6}$$

Here, $f^e, g^e$ denote edge-update networks (MLPs applied on edge features), while $f^n, g^n$ denote node-update networks. This design establishes a theoretically exact reversible process in the latent space, ensuring consistent bidirectional rollouts of graph-structured particle systems while scaling to large node sets and complex interactions.

In practice, RRMP scales to thousands of particles and complex interactions while retaining theoretical guarantees of invertibility. The combination of residual coupling, message passing, and recomputed interaction graphs makes RRMP both expressive and stable, ensuring that forward and backward simulations remain tightly aligned.

## 4 EXPERIMENTS

### 4.1 EXPERIMENT SETTINGS

**Dataset.** We evaluate on three dissipative fluid datasets of increasing complexity: *WaterDrop*, *WaterRamp*, and *Water_3D* (Sanchez-Gonzalez et al., 2020). *WaterDrop* (2D) contains up to 1k particles with simple droplet dynamics. *WaterRamp* (2D) increases difficulty with up to 2.3k particles interacting with complex obstacles. *Water_3D* is a large-scale setting with rich fluid motions and up to 13k particles. This progression from small-scale to large-scale systems provides a rigorous testbed for evaluating model generalization and scalability.

**Task & Baseline.** To comprehensively assess R-GNS, we consider three tasks: (1) **Forward Simulation**. Given initial states, the model predicts long-horizon rollouts (300 steps) and is compared against leading neural feed-forward simulators, including GNS (Sanchez-Gonzalez et al., 2020), DMCF (Prantl et al., 2022), EGNN (Satorras et al., 2021; Toshev et al., 2023), and NeuralSPH (Toshev et al., 2024). (2) **Inverse Inference**. The goal is to recover initial states from given outcomes. We benchmark R-GNS against optimization-based solvers with differentiable framework DiffTaichi, physics solver SPH and the Adam optimizer (Hu et al., 2019; Li et al., 2023b). For neural baselines, we compare with two separately trained GNS models, denoted as Sep-GNS. To probe stability, we evaluate prediction consistency under varying numbers of backward steps, in addition to efficiency. (3) **Goal-Conditioned Tasks**. To demonstrate full bidirectional reasoning, we task simulators with generating physically plausible trajectories that guide fluid waves into target shapes ("L", "N").

**Evaluation Metrics.** For forward quality, we use Rollout MSE, mean squared error between predicted and ground-truth rollouts, as the primary metric, with additional distributional metrics OT, MMD (Villani et al., 2008; Gretton et al., 2012) reported in Appendix. For inverse and goal-conditioned performance, we report Consistency MSE, measuring temporal coherence and forward–inverse alignment. Efficiency is evaluated by inference time, memory and parameter counts.

**Implementation Details.** We implement the Reversible Residual Message-Passing network with 10 propagation steps and a hidden dimension of 128. Within reversible blocks, the feature vector is split into two 64-dimensional halves for coupling updates. A linear projection with pseudo-inverse is used as the encoder–decoder, and the dynamic quantity is masked as the only predicted output to preserve symmetry. All models are trained using the Adam optimizer with an initial learning rate of 1e-4 and cosine decay. The batch size is set to 2, and early stopping is applied based on validation loss. For scalability, training is conducted on NVIDIA A800 GPUs, with smaller datasets (*WaterDrop*, *WaterRamps*) using 2 GPUs and the larger *Water_3D* dataset using 4 GPUs.

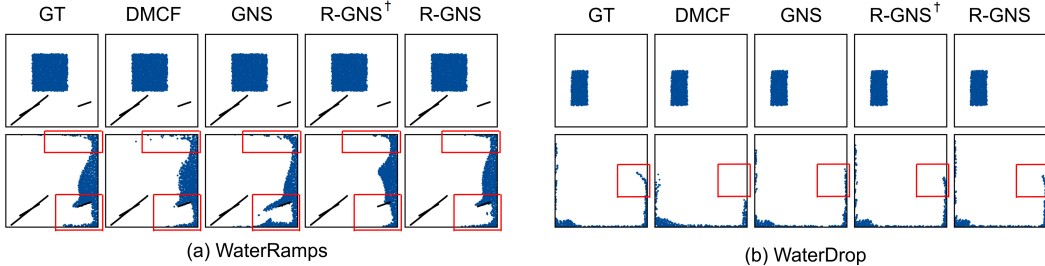

(a) WaterRamps                    (b) WaterDrop

Figure 3: Qualitative results on 2D datasets. R-GNS is able to achieve more faithful results on 2D datasets. On *WaterDrop*, R-GNS more accurately reproduces the intricate structures of wave motion during simulation. On *WaterRamps*, R-GNS more accurately captures the motion patterns of fluids in environments with complex fluid–obstacle interactions.

## 4.2 RESULTS

### 4.2.1 FORWARD SIMULATION

We first evaluate forward prediction accuracy across all three datasets. **Graph Network vs. Convolutional Network.** Qualitative results on the 2D datasets (*WaterDrop* and *WaterRamp*) are presented in Fig. 3, covering the convolutional method DMCF, the strongest graph-based baseline (GNS), our bidirectional R-GNS, and its unidirectionally trained variant (R-GNS†). In comparison with the convolutional baseline, graph-based methods more effectively capture fine-grained particle interactions, reflecting their suitability for particle dynamics. Within this category, R-GNS most faithfully reproduces intricate wave structures in the *WaterDrop* scene and demonstrates strong capability in modeling fluid–obstacle interactions in the *WaterRamp* scenario.

**Prediction Accuracy Comparison.** Full quantitative comparisons with all methods are reported in Table 1. R-GNS consistently achieves the lowest rollout MSE across all datasets, with additional metrics (MMD and OT) in the Appendix confirming the same trend. Across datasets, R-GNS consistently achieves the best forward accuracy among all baselines. R-GNS† generally outperforms GNS, though the margin varies across datasets. Moreover, R-GNS outperforms all other non-reversible neural baselines, demonstrating the broad benefit of unifying bidirectional simulation. This highlights that a unified reversible framework yields improved forward accuracy through richer bidirectional supervision.

Table 1: Rollout MSE (1e-3) in the Forward Process

| Dataset/Methods | NeuralSPH | EGNN | DMCF | GNS | R-GNS† | R-GNS |
|---|---|---|---|---|---|---|
| Water_3D | 14.2 | 15.2 | 26.0 | 12.1 | 13.8 | **9.5** |
| WaterDrop | 0.67 | 1.41 | 3.20 | 0.70 | 0.59 | **0.31** |
| WaterRamp | 10.5 | 15.9 | 16.3 | 10.9 | 10.1 | **9.0** |

**Efficiency Comparison.** Table 2 reports efficiency metrics across models. R-GNS requires only one quarter of the parameters of Sep-GNS (i.e., half of a single GNS) and substantially reduces inference memory. Its inference speed is comparable to GNS, the fastest baseline. Overall, R-GNS demonstrates markedly higher efficiency.

Table 2: Efficiency Comparison on the WaterDrop Dataset

| Metrics / Methods | NeuralSPH | EGNN | DMCF | GNS | R-GNS† | R-GNS |
|---|---|---|---|---|---|---|
| Parameter Count (M) | 1.72 | 4.27 | 2.17 | 1.59 | 0.78 | **0.78** |
| Inference Time (ms/step) | 14.52 | 34.01 | 14.18 | **11.27** | 13.41 | 13.41 |
| Inference Memory (MB) | 103.64 | 319.52 | 343.11 | 91.55 | 59.27 | **59.27** |

Taken together, these results demonstrate that R-GNS delivers the best overall trade-off: it achieves state-of-the-art accuracy in forward simulation while maintaining superior efficiency in terms of parameters, memory, and inference speed.

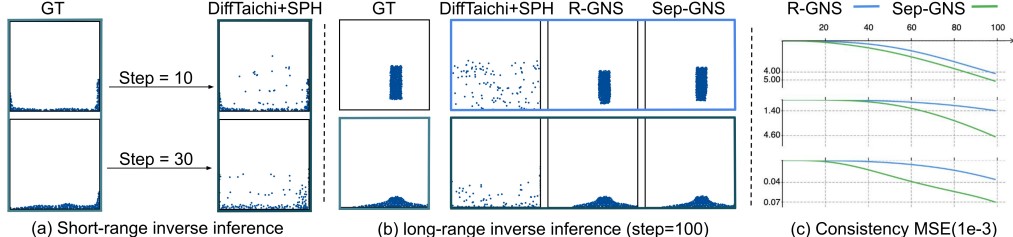

(a) Short-range inverse inference     (b) long-range inverse inference (step=100)     (c) Consistency MSE(1e-3)

Figure 4: Inverse inference results. (a) Short-range inference: optimization-based solvers are unstable. At step 10 they roughly recover the fluid shape but already exhibit noticeable noise, which further amplifies by step 30, leading to inaccurate solutions. (b) Long-range inference at step 100: optimization completely collapses, whereas feed-forward approaches (R-GNS and Sep-GNS) maintain stable and accurate predictions. R-GNS achieves more faithful recovery, including more accurate initial heights and forward simulation. (c) Inverse–forward consistency across varying steps: both feed-forward models converge toward near-zero error within 40 steps, with R-GNS consistently achieving higher consistency than Sep-GNS over the entire range.

### 4.2.2 INVERSE INFERENCE

**Inverse Solve Stability.** Inverse inference results are shown in Fig. 4(a–b). (a) Optimization-based solvers exhibit unstable behavior: even on the simpler *WaterDrop* dataset, short-range inference already introduces noticeable noise, and solutions quickly deteriorate. On the more challenging *Water-3D* dataset, where particle counts are an order of magnitude larger, we observed that optimization-based solvers collapse after as few as 5 steps, producing no valid solutions. (b) In contrast, neural feed-forward methods demonstrate far greater stability. On *WaterDrop*, they maintain coherent trajectories over long horizons, while optimization-based solvers collapse. Among them, R-GNS achieves the most reliable rollouts, preserving physically plausible dynamics even at long-range inference.

**Consistency Comparison.** As discussed earlier, optimization-based solvers deteriorate rapidly, exceeding $10^{-2}$ consistency error within fewer than 10 steps. In contrast, the neural approaches compared in Fig. 4(c) maintain errors within the $10^{-3}$ scale even at 100 steps. R-GNS further improves over Sep-GNS, exhibiting higher consistency and slower error growth, maintaining reliable forward–inverse alignment over long horizons.

**Comprehensive Comparison.** Table 3 reports consistency MSE, inference time, parameter counts, and maximum stable step. R-GNS runs over 100× faster than optimization-based solvers, achieves longer stable rollouts, and uses only one quarter the parameters of Sep-GNS while maintaining higher consistency. These advantages make it particularly suitable for large-scale dissipative fluid simulations.

Table 3: Inverse Inference Comparison on the WaterDrop (inverse step=40)

| Methods | | Optimization Method | Feed-Forward Methods | |
|---|---|---|---|---|
| | | DiffTaichi+SPH | Sep-GNS | R-GNS |
| Consistency MSE | (1e-3) | 9.07 | 0.017 | **0.002** |
| Inference Time | (s) | 236.792 | **0.451** | 0.536 |
| Parameter Counts | (M) | / | 3.18 | **0.78** |
| Max Stable Steps | | <40 | >100 | **>100** |

In summary, R-GNS delivers stable and consistent inverse inference across diverse datasets, sustaining reliable forward–inverse alignment even under long rollouts. Especially in dissipative fluids, where inversion is inherently ill-posed and optimization-based solvers break down, R-GNS exploits forward dynamics as a physical prior, enabling accurate and efficient recovery of initial states.

### 4.2.3 GOAL-CONDITIONED TASK

The *Water-3D* dataset, with thousands of particles and dissipative dynamics, is intractable for optimization-based solvers, which collapse after only a few steps. As shown in Fig. 5, R-GNS

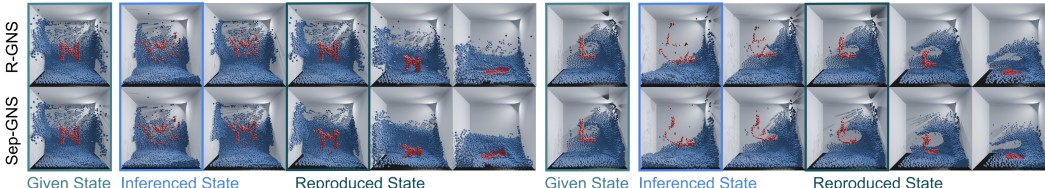

Figure 5: Goal-conditioned simulation on the *Water_3D* dataset. Target particle configurations shaped as "L" and "N" (given state) are inverse-inferenced to plausible initial states (inferenced state) and then rolled forward to reproduce the targets (reproduced state). R-GNS achieves close reproduction of the intended shapes, whereas Sep-GNS loses structural fidelity.

successfully addresses this task through consistent bidirectional inference and forward simulation. It reliably scales to large particle counts and reproduces target shapes with high fidelity, whereas Sep-GNS fails to maintain structural consistency. These results highlight the robustness of R-GNS's unified reversible framework, leveraging forward physical priors to enable reliable inverse inference even in dissipative fluid systems.

### 4.3 ABLATION STUDY

Fig. 6 summarizes the ablation results. (a) Compared four loss designs: DL_1 (MSE and MMD), DL_2 (MSE, cosine similarity and MMD), DL_3 (MSE and cosine), DL_4 (MSE). Using only MSE (DL_4) yields the best performance in both forward accuracy and consistency, suggesting auxiliary losses may introduce noise. (b) ILP, as a reversible encoder–decoder, substantially improves inverse inference consistency compared to MLP. (c) Fixing edge features is not critical but provides a modest performance gain, supporting its inclusion in our final design.

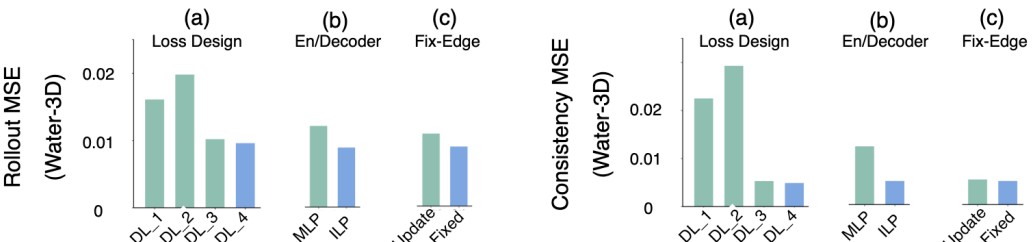

Figure 6: Ablation study on the *Water-3D* dataset, measuring rollout and consistency MSE. (a) Four Loss designs (b) Encoder–decoder: MLP vs. ILP. (c) Edge features: fixed vs. update in latent space.

## 5 CONCLUSION & FUTURE WORK

We introduced R-GNS, the first reversible simulator for dissipative fluids. As a bidirectional simulator, it achieves inverse inference at the same fast speed as forward rollout, orders of magnitude faster than optimization-based solvers. Built on a unified reversible framework that couples forward and inverse simulation, R-GNS attains higher accuracy with only one quarter of the parameters of separate models. Its reversible components (ILP and RRMP) ensure mathematically exact propagation and strict forward–inverse consistency, while shared parameters and bidirectional training leverage forward physical priors to capture underlying laws and recover faithful initial states, even in dissipative systems with multiple plausible solutions.

R-GNS enables efficient goal-conditioned control of dissipative fluids. For example, it can drive flows to form user-defined shapes, making it useful for controllable simulation in world models, embodied agents, and visual effects. However, Richer goal conditioning demands finer fluid detail, implying much larger particle counts (on the order of $10^5$–$10^6$ or more). Although R-GNS supports long-horizon rollouts at about $10^4$ particles, computational efficiency degrades as system size grows; scaling further will likely require techniques such as hierarchical message passing, domain decomposition, and multi-resolution scheduling, which we leave for future exploration.

## 6 ETHICS STATEMENT

This work does not involve human subjects, sensitive personal data, or content that raises direct ethical concerns such as discrimination, bias, or privacy violations. All datasets used (*WaterDrop*, *WaterRamp*, and *Water-3D*) are particle-based fluid simulations, and thus do not introduce risks related to privacy or consent. The proposed methodology is intended for advancing physically consistent simulation research, with potential applications in scientific computing, graphics, and robotics. While faster and more accurate fluid simulation may also be used in entertainment and design applications, we do not foresee harmful dual-use scenarios. All authors have read and adhered to the ICLR Code of Ethics, and confirm that this submission complies with the principles of research integrity, transparency, and reproducibility.

## 7 REPRODUCIBILITY

We have structured the paper to facilitate reproducibility. **Model architecture** and **implementation details**, including full hyperparameter settings, are provided in Sec. 3.3 and Sec. 4.1. **Theoretical guarantees of reversibility** are formally stated in the Appendix A. **Datasets**, **evaluation protocols**, and **baseline configurations** are described in Sec. 4.1.

To further support reproducibility, demo videos are included in the supplementary materials, and the full source code will be released with the camera-ready version.

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

## A    METHODOLOGY PROOFS

In this section, we present the theoretical justification and implementation details of the two core reversible components: ILP encoder-decoder and the Residual Reversible Message-Passing Network.

### A.0.1    PROOF OF ILP ENCODER-DECODER

We employ a single-layer linear network as the encoder and its corresponding inverse transformation as the decoder. This process is formalized in Equation 3. The encoding matrix $W \in R^{b \times a}$ projects node features from physical space $R^a$ to latent space $R^b$ (where $b > a$). To enable reversibility, we compute its pseudoinverse $W^{\dagger}$ via singular value decomposition (SVD), which serves as the decoding transformation. The computation of the pseudoinverse proceeds as follows:

First, compute the singular value decomposition (SVD) of $W$, expressed as:

where $r = \mathrm{rank}(W)$, $U \in \mathbb{R}^{a \times a}$ and $V \in \mathbb{R}^{b \times b}$ are orthogonal matrices, $V^T$ denotes the transposes of matrix $V$. $\Sigma$ is the singular value matrix, with non-negative singular values along its main diagonal, $\Sigma = \mathrm{diag}(\sigma_1, \ldots, \sigma_r, 0, \ldots, 0)$.

The pseudoinverse is then given by:

$$W^{\dagger} = V \Sigma^{\dagger} U^{\top}, \quad \Sigma^{\dagger} = \mathrm{diag}\left(\sigma_1^{-1}, \ldots, \sigma_r^{-1}, 0, \ldots, 0\right) \tag{7}$$

Since $W$ is a non-square matrix, decoding via its pseudoinverse corresponds to the least-squares solution of the inverse encoding process, that is:

$$\tilde{\chi} = \arg\min_{\chi} ||W\chi - (n - B)||_2^2 \tag{8}$$

We quantified the reconstruction error of this process, finding that $||\chi - \mathrm{dec}(\mathrm{enc}(\chi))||^2 < 10^{-6}$. Therefore, the encoder-decoder constructed via pseudoinverse and matrix transformations demonstrates nearly perfect reversibility.

### A.0.2    PROOF OF RESIDUAL REVERSIBLE MESSAGE-PASSING NETWORK

We propose a Residual Reversible Message-Passing Network that performs forward or backward propagation of node features in the latent space, enabling an invertible mapping between node states at time steps $t$ and $t + 1$. For a Residual Reversible Message-Passing Network with $M$ layers, the forward and backward propagation between layer $l$ and layer $l + 1$ are formally defined in Equations 5 and 6. At each propagation step, the features of all nodes in the graph are simultaneously updated. The forward and backward processes can be compactly expressed as:

$$\{n^{l+1}\}_{i=1}^N = \mathrm{layer}_{fwd}^{l+1}(\{n^l\}_{i=1}^N, \{e\}) \tag{9}$$

and

$$\{n^l\}_{i=1}^N = \mathrm{layer}_{bwd}^{l+1}(\{n^{l+1}\}_{i=1}^N, \{e\}) \tag{10}$$

The full forward propagation over $M$ layers is obtained by iteratively applying Equation 9 for $M$ steps:

$$\begin{cases} \{n^1\}_{i=1}^N = \mathrm{layer}_{fwd}^1(\{n^0\}_{i=1}^N, \{e\}) \\ \{n^2\}_{i=1}^N = \mathrm{layer}_{fwd}^2(\{n^1\}_{i=1}^N, \{e\}) \\ \ldots \\ \{n^M\}_{i=1}^N = \mathrm{layer}_{fwd}^M(\{n^{M-1}\}_{i=1}^N, \{e\}) \end{cases} \tag{11}$$

The corresponding full reverse propagation is performed by applying Equation 10 in reverse order for $M$ steps:

$$\begin{cases} \{n^{M-1}\}_{i=1}^{N} = \text{layer}_{fwd}^{M}(\{n^{M}\}_{i=1}^{N}, \{e\}) \\ \{n^{M-2}\}_{i=1}^{N} = \text{layer}_{fwd}^{M-1}(\{n^{M-1}\}_{i=1}^{N}, \{e\}) \\ ... \\ \{n^{0}\}_{i=1}^{N} = \text{layer}_{fwd}^{1}(\{n^{1}\}_{i=1}^{N}, \{e\}) \end{cases} \tag{12}$$

## B  ADDITIONAL EXPERIMENTS

### B.1  EVALUATION WITH ADDITIONAL METRICS

To provide a more comprehensive assessment of simulation accuracy, we report results using two additional metrics. **Optimal Transport (OT)** quantifies global differences in particle distributions, while **Maximum Mean Discrepancy (MMD)** captures discrepancies in high-dimensional feature space. As summarized in Table 4, R-GNS consistently achieves the best performance across both metrics, further demonstrating its superior accuracy and stronger adherence to physical consistency.

Table 4: Simulation Performance Across Models on WaterDrop

| Methods | NeuralSPH | EGNN | DMCF | GNS | R-GNS* | R-GNS |
|---|---|---|---|---|---|---|
| OT (1e-3) | 0.31 | 0.49 | 1.37 | 0.27 | 0.25 | **0.18** |
| MMD (1e-2) | 0.52 | 0.79 | 0.82 | 0.60 | 0.57 | **0.23** |

## C  USE OF LARGE LANGUAGE MODELS (LLMs)

We used Large Language Models (LLMs) as a general-purpose writing assistant. Their role was limited to grammar correction, wording refinement, and improving clarity of exposition. All research ideas, technical methods, experiments, and results were conceived, implemented, and validated entirely by the authors.

The LLM did not contribute to research ideation, design of experiments, or interpretation of results, and thus cannot be considered a scientific contributor. All substantive content of the paper remains the sole responsibility of the authors.

