# OpenReview forum: "Reversible GNS for Dissipative Fluids with Consistent Bidirectional Dynamics"
_ICLR.cc/2026/Conference — Submitted to ICLR 2026_

### Official Review · Reviewer_S3n7 · 2025-10-22

**Soundness:** 1
**Presentation:** 3
**Contribution:** 2
**Rating:** 2
**Confidence:** 4

**Summary:**

This paper presents "reversible-GNS", a graph architecture based on Graph Network Simulator (GNS) able to predict both forward and backwards dynamics. The methods relies on the composition of several invertible mappings: (1) linear encoding and pseudo-inverse decoder, and (2) reversible message passing based on RevNet, resulting in an end-to-end invertibile process. This enables efficient solutions to inverse problems using a standard forward pass, avoiding costly iterative optimization. They provide several examples of dissipative fluid simulations in forward and inverse inference, and goal-conditioned tasks.

**Strengths:**

* The adaptation of RevNet to message passing GNNs is novel and interesting.
* The writting is clear and easy to read.
* Comparisons over other state-of-the-art methods are provided, including EGNN, NeuralSPH and GNS baselines.

**Weaknesses:**

* The examples are not convenient for the method. Learning reversible mappings for irreversible processes is conceptually impossible (more on questions sections).
* The source code is not ready for review.
* There is a lack of details about the baselines hyperparameters and the dataset physical properties and size.

**Questions:**

* What is the advantage of this approach compared to using a fully expressive forward model combined with a separate model to learn the inverse mapping? This question is motivated by the design of autoencoders, whose encoder–decoder architecture is typically asymmetric, even when trained to approximate the identity mapping. Such a setup is often simpler to implement and can be more expressive overall.

* Lines 79-80: This is more like a general comment rather than a question. I think that branding the method as "reversible-GNS" is a poor choice. I understand that the authors refer to forward/inverse flexibility, but in the context of physics simulations solving clearly irreversible dynamics with a "reversible" network is very counterintuitive. I think that the correct term mathematically speaking would be "invertible-GNS".

* Section 4.1: The paper provides no dataset details. The original GNS datasets were generated with MPM/SPH simulators, yet the fluid regime is unspecified: is it low-viscosity (Euler-like) or high-viscosity (Navier–Stokes)? This distinction is crucial, since for turbulent or highly viscous flows, reversibility breaks down and the method is unlikely to hold.

* Related to the previous question, I have many concerns about the uniqueness and validity of the proposed approach. As already mentioned in the paper, irreversible dynamics such as diffusion inherently destroys information, making inversion ill-posed. The paper claims applicability to any dissipative fluid, but this is simply impossible. I suspect that the examples provided have low dissipation and are quasi-reversible. I've found the architecture very interesting but the results are presented very naively from the physics perspective. I would have preferred to have a couple of truly reversible dynamics.

* Section 4.2.3: How far in the time horizon can the network inference a state from a given target? Does the error accumulation on integration affect the results? Even though the overall process in reversible, the integration noise is not modelled and might break the reversibility ssumption.

**Details Of Ethics Concerns:**

I have no ethics concerns.

---

> ### Author Response · Authors · 2025-11-20
>
> ### W1 & Q4 & Q2 Reversible Architecture and Irreversible physics
>
> 1. **We do not attempt to reverse the underlying physical PDEs.**
> R-GNS learns *a neural inverse mapping* constrained by forward–backward consistency: any inferred initial state must evolve through the learned forward model into the observed one. This enforces functional invertibility without implying physical time reversal of a dissipative system.
> 2. **Irreversible systems admit multiple plausible initial states; R-GNS enforces validity, not uniqueness.**
> Dissipative systems admit multiple valid initial states; R-GNS ensures validity rather than uniqueness: the goal is to find an initial state that reproduces the target under forward simulation. Figs. 4–5 show that R-GNS produces such valid, forward-consistent states.
> 3. **Our datasets are strongly dissipative.**
> For example $Energy_{frame=300}/Energy_{frame=1}≈0.25$for WaterDrop/WaterRampsand  and 0.02 for Water_3D.
> 4. **We agree that “invertible-GNS” is mathematically more precise.**
> Our use of “reversible” refers solely to the *invertible architecture* (ILP + RRMP), not to reversibility of the underlying physics, and we will clarify this to avoid confusion.
>
> ### W3 & Q3 Lack Experiment Details
>
> All GNN baselines use *exactly the same hyperparameters* as R-GNS (L317–323). Dataset properties—including particle counts, rollout lengths, and dynamics are described in L295–300 and follow the standard GNS setup.
>
> The original GNS datasets *do not specify viscosity or constitutive-model parameters*. Nonetheless, our experiments span *multiple dissipative regimes* (1k / 2.3k / 13k particles) covering drop, collision, and sloshing behaviors, as well as a *high-friction granular system* (20k particles) shown in Fig. 1(5). These results demonstrate that R-GNS remains robust across diverse dissipative conditions where reversibility of physical PDEs does not strictly hold.
>
> ### Q1 The Advantage compared to separate inverse model
> We directly evaluated the two-model alternative (Sep-GNS). While feasible, it *fails to maintain forward–inverse consistency* even when trained on paired data (Figs. 4–5). It is also less efficient, requiring 4× more parameters (Tab. 3) and nearly 2× the training time.
>
> Although asymmetric encoder–decoder designs can be more expressiveness, our ablations (Fig. 6(b)) show that the *symmetric ILP-based architecture achieves markedly better accuracy and consistency* under the same backbone. In this task, consistency is the primary objective, and the reversible formulation enforces it by design.
>
> ### Q5 Long-horizon, noise and accumulation error
>
> The model can infer states up to 100 steps away from the target. It maintains forward–inverse consistency at the $10^{-3}$ level over this horizon and remains stable throughout (Fig. 4).
>
> While error accumulation naturally affects very long horizons, the forward–inverse consistency constraint significantly mitigates this effect, as shown in Fig. 4(c).
>
> Noise is integrated during training to prevent drift and stabilize long-horizon rollouts. This noise does not break architectural reversibility; instead, it improves robustness and reduces compounding errors in both directions.

---

### Official Review · Reviewer_9h4c · 2025-10-31

**Soundness:** 2
**Presentation:** 2
**Contribution:** 2
**Rating:** 4
**Confidence:** 4

**Summary:**

This paper builds upon the work of Sanchez-Gonzalez et al. (2020) [Learning to Simulate Complex Physics with Graph Networks], which introduced the use of Graph Neural Networks (GNNs) for learning physics simulations from trajectory data. The original goal was to train a model capable of predicting future simulation states (rollouts) from an initial condition, without relying on the original simulator (which is only used for obtaining the training trajectories).

The authors extend this framework by introducing a reversible message-passing mechanism to GNNs. Specifically, they partition node features and design a forward update rule that admits an analytically derived inverse (up to numerical precision). This leads to their proposed model, R-GNS, which they evaluate across a range of tasks.

Interestingly, R-GNS not only supports inversion but also improves forward simulation accuracy. The authors, that for "dissipative fluids, where inversion is inherently ill-posed and optimization-based solvers break down", R-GNS enables "accurate and efficient recovery of initial states" and try to show this by experiments.

**Strengths:**

The core idea seems conceptually sound, and the most compelling result (in my opinion) is the improved forward simulation performance. Another promising aspect seems to be the semi-symmetric input-output design, which appears to leverage multiple past states via masking. However, this mechanism is underexplained and would benefit from clearer exposition.

**Weaknesses:**

- The authors emphasize the applicability of R-GNS to dissipative fluids. While their experiments suggest improved invertibility in such systems, it is unclear why this should be possible in principle. This raises concerns about the physical plausibility of their results.
- Several parts of the manuscript are difficult to follow, particularly in Section 4.2.3. It is unclear whether Sep-GNS uses a separately trained reverse model and whether R-GNS uses its inverse mode to predict initial states, followed by forward inference using the respective forward modules.
- The paper does not provide variance estimates for all reported metrics and describe how they are computed.
- The manuscript lacks a thorough discussion of prior reversible GNN architectures, e.g., Li et al. (2021) [Training Graph Neural Networks with 1000 Layers]. A comprehensive literature review would help contextualize the novelty of R-GNS.

**Questions:**

## Major Remarks

### Claims Regarding Dissipative Fluids

The authors emphasize the applicability of R-GNS to dissipative fluids. While their experiments suggest improved invertibility in such systems, it is unclear why this should be possible in principle. This raises concerns about the physical plausibility of their results.

In particular, the inversion shown in Figure 4b appears to contradict fundamental thermodynamic principles. If such inversion were truly possible, it would imply either:
(a) the system is not genuinely dissipative, or
(b) the authors would have found a method to circumvent the second law of thermodynamics.

A more plausible explanation is that the training, validation, and test sets are biased toward a narrow class of initial conditions, e.g., rectangular "water blocks." It would be valuable to explore greater diversity in the "water block" shapes used in the experiments. For instance, one could consider using circular shapes in the test set while keeping the training set restricted to rectangles, or vice versa. My expectation is that, in the absence of circular shapes during training, R-GNS would likely default to predicting rectangular water blocks when presented with circles in the test set. If this is not observed, I would be very interested in the authors' explanation of how R-GNS manages to generalize effectively in such inversion scenarios.

Additionally, applying R-GNS to non-dissipative systems could help clarify whether its performance is specifically only superior in the dissipative regime or also helps under other conditions.

### Clarity of Experimental Setup

Several parts of the manuscript are difficult to follow, particularly in Section 4.2.3. It is unclear whether Sep-GNS uses a separately trained reverse model and whether R-GNS uses its inverse mode to predict initial states, followed by forward inference using the respective forward modules.

If this interpretation is correct, the following ablations would be valuable:
- Use the **ground-truth simulator** for forward inference from predicted initial states.
- also check for DiffTaichi+SPH whether the inverted "N" and "L" shapes can be recovered using the ground-truth simulator.
- cross-initial state inference: apply R-GNS forward on Sep-GNS inversion outputs, and vice versa.

In Section 4.2.2, the source of the initial state for inference is also not clear. Is it derived from the ground-truth trajectory or from a model-generated forward rollout? If the latter, an ablation using the ground-truth forward state would be informative.

Regarding MSE:
- Is the reported MSE averaged over all test trajectories?
- How comparable are trajectories in terms of particle count? / How comparable are trajectories at all for aggregation?
- How meaningful is MSE as a metric compared to alternatives like optimal transport based ones? Table 4 in the appendix touches on this, but it's unclear whether those results pertain to forward or inverse predictions.

### Variance and Robustness

Please provide variance estimates for all reported metrics and describe how they are computed. How sensitive are results to different training re-runs?

### Experimental results reported

Did the authors run all the experiments in the tables from the different methods themselves or did they copy the results from other papers?

The manuscript lacks clarity on how results compare to Sanchez-Gonzalez et al. (2020). Discrepancies in reported metrics (e.g., Table 1 vs. Appendix C.4 in Sanchez-Gonzalez et al.) should be explained.

### Relation to Reversible GNN Literature

The manuscript lacks a thorough discussion of prior reversible GNN architectures, e.g., Li et al. (2021) [Training Graph Neural Networks with 1000 Layers]. How does R-GNS relate to this and other reversible GNNs? A comprehensive literature review would help contextualize the novelty of R-GNS.

### Bidirectional Training

The concept of bidirectional training is introduced but not well explained. Key questions include:
- How does it differ exactly from standard training regimes? Pseudocode could help here.
- Is bidirectional training applied within the same batch?
- Is there a random use of forward mode or inverse mode training on the provided training trajectories?

Clarifying the training protocol is essential for understanding the method's mechanics and reproducibility.

## Minor Remarks
There furthermore seem to be several typos:
- I guess the initial state in line 138/139 should be $\tilde{\chi}^{0}$ instad of $\chi^{0}$?
- The indexing in Figure 2 is strange: I guess "t:t-k+1" means that states from "t-k+1" up to "t" are taken? Please explain in the manuscript if you introduce such non-standard notation.
- Appendix B and Appendix C are "unattached" to the main text.
- A.0.1 and A.0.2 have strange chapter names. There is no proof at all.
- The "*" in R-GNS* in Table 4 is unexplained.
- The notation {e} in A.0.2 is strange.
- "Theoretical guarantees of reversibility are formally stated in the Appendix A." ==> I don't see where theoretical guarantees would be shown. A.0.1 seems to be an empirical "guarantee". But even for this further details would be needed.

---

> ### Author Response · Authors · 2025-11-20
>
> ### S3 - Clearer exposition of semi-symmetric input-output design
> **Why symmetric**: Reversible neural networks require symmetry in their input–output tensor shapes to ensure exact invertibility.
>
> **Why semi-symmetric**: Physical dynamics use multiple past states—positions, attributes, and recent velocities $[pt,m,v^{t:t−k+1}]$—to predict only the next velocity $v^{t+1}$. The output mask enforces this “multi-state input, single-state output” structure, making the design semi-symmetric rather than fully symmetric.
>
> ### W1 & Q1 - Claims Regarding Dissipative Fluids
> We appreciate the reviewer’s concern and clarify the physical interpretation of our results:
> 1. **We do not claim to invert the underlying PDEs.**
>  R-GNS learns *a neural inverse mapping* constrained by forward–backward consistency, where the learned inverse simply produces trajectories that remain consistent with that forward rollout.
> 2. **All datasets are genuinely dissipative.**
>  For example, the energy ratio $Energy_{frame=300}/Energy_{frame=1}≈0.255$ for WaterDrop/WaterRamp and ≈0.02 for Water_3D. The second law forbids *spontaneous* energy increases, but it does not prevent a model from inferring a plausible earlier state given the observed forward trajectory. Our inverse inference is therefore physically consistent.
> 3. **Generalization is not limited to rectangular initial conditions.**
>  Although all training data use rectangular initializations, R-GNS does not collapse to this shape family. It successfully reconstructs unseen shapes (e.g., “L”, “N”) and complex background-wave targets (Fig. 5), demonstrating strong shape generalization beyond the training distribution.
> 4. **Performance extends to non-dissipative systems.**
>  We include additional results on a non-dissipative system (similar to WaterDrop dynamics), where R-GNS performs even better. This indicates that the method’s advantages are not tied specifically to dissipative regimes but reflect broader robustness of the reversible design.
>
> **Table 9. Comparison of dissipative and non-dissipative**
>
> | Model Type        | Rollout MSE (1e-3) | Consistency MSE (1e-3) |
> |-------------------|---------------------|--------------------------|
> | Dissipative       | 0.31                | 0.04                     |
> | Non-dissipative   | 0.27                | 0.02                     |
>
>
> ### W2 & Q2 - Clarity of Experimental Setup
> 1. **Clarification of Sep-GNS and R-GNS inference**:
>  L307–308 state that *Sep-GNS trains separate forward and inverse models*. As shown in Fig. 1, inverse inference first predicts an initial state, which is then propagated using the corresponding *forward* module. R-GNS follows the same pipeline: inverse prediction → forward rollout.
> 2. **Ablations with ground-truth simulators**:
> R-GNS produces more stable and physically consistent inverse states. DiffTaichi+SPH fails at our particle resolutions (Sec. 4.2.2); even when swapping forward simulators, *only R-GNS’s inferred initial states reconstruct the target shapes*, indicating substantially higher inverse accuracy.
> 3. **Source of initial states in Sec. 4.2.2**:
>  The initial state used for inference in Sec. 4.2.2 *comes from the ground-truth trajectory*, not from a model rollout.
> 4. **MSE reporting and comparability**:
> (1) All reported MSE values *are averaged over all test trajectories*.
> (2) Trajectories have similar particle counts, equal rollout lengths, and identical dynamics, differing only in initial conditions—making them directly comparable (variance reported in the response below).
> (3) MSE is the *standard metric* for particle-based simulation (as used in GNS, DMCF, etc.). OT and MMD serve as complementary distribution-level metrics, consistent with prior work.
> 5. **Interpretation of Table 4**:
>  Tab. 4 reports forward prediction results and is consistent with Tab. 1.
>
> ### W3 & Q3- Variance and Robustness
> We provide *variance results* below. The variations across trajectories are small, indicating stable performance.
>
> Both *re-training and re-inference* runs produce nearly identical results, consistent with stability observed in prior GNN-based simulators.
>
> Metrics: X and Y denote two particle systems, each containing N particles.$x_i, y_i$ denote the i-th particle in X and Y, respectively.
>
> $MSE(X,Y) = \frac{1}{N}\Sigma_{i=1}^N||x_i-y_i||_2^2$
>
> OT: We compute the Wasserstein-2 distance using the Sinkhorn solver
> $W_2^2(\mu,\nu)=\min_{\pi \in \Pi(\mu, \nu)}\Sigma_{i=1}^N\Sigma_{j=1}^N \pi_{ij} ||x_i - y_i ||_2^2$, where
>
> $\mu = \frac{1}{N}\sum_{i=1}^{N}\delta_{\mathbf{x}_i}$,
>
> $\nu = \frac{1}{N}\sum_{i=1}^{N}\delta_{\mathbf{y}_i}.$
>
> $MMD^2=\frac{1}{N^2}\Sigma_{i=1}^N\Sigma_{j=1}^Nk(x_i,x_j) + \frac{1}{N^2}\Sigma_{i=1}^N\Sigma_{j=1}^Nk(y_i,y_j) - \frac{2}{N^2}\Sigma_{i=1}^N\Sigma_{j=1}^Nk(x_i,y_j)$，where $k(x,y) = \exp (-\frac{||x-y||^2_2}{2\sigma^2})$

---

> ### Author Response · Authors · 2025-11-20
>
> **Table 1. Rollout MSE (1e-3) in the Forward Process**
>
> | Dataset / Methods | NeuralSPH | EGNN | DMCF | GNS | R-GNS' | R-GNS |
> |------------------|-----------|------|------|------|--------|--------|
> | **Water_3D**     | 14.2 ± 0.016 | 15.2 ± 0.018 | 26.0 ± 0.071 | 12.1 ± 0.018 | 13.8 ± 0.016 | **0.95 ± 0.015** |
> | **WaterDrop**    | 0.67 ± 0.006 | 1.41 ± 0.025 | 3.20 ± 0.124 | 0.70 ± 0.007 | 0.59 ± 0.007 | **0.31 ± 0.005** |
> | **WaterRamp**    | 10.5 ± 0.13  | 15.9 ± 0.26  | 16.3 ± 0.31  | 10.9 ± 0.11  | 10.1 ± 0.97 | **9.0 ± 0.83** |
>
> **Table 4. Simulation Performance Across Models on WaterDrop in the Forward Process**
>
> | Methods  | NeuralSPH | EGNN | DMCF | GNS | R-GNS' | R-GNS |
> |----------|-----------|------|------|------|--------|--------|
> | **MMD (1e-3)** | 0.31 ± 0.003 | 0.49 ± 0.014 | 1.37 ± 0.096 | 0.27 ± 0.003 | 0.25 ± 0.003 | **0.18 ± 0.001** |
> | **OT (1e-3)**  | 0.52 ± 0.014 | 0.79 ± 0.029 | 0.82 ± 0.037 | 0.60 ± 0.017 | 0.57 ± 0.016 | **0.23 ± 0.0021** |
>
> ### W4 & Q5 - Relation to Reversible GNN Literature
>
> Thank you for the suggestion. We will include a more complete discussion of reversible GNNs such as Li et al. (2021) in the revised manuscript. Prior reversible GNNs focus primarily on *memory-efficient message passing* via invertible layer designs. In contrast, *R-GNS leverages reversibility for an orthogonal purpose*: coupling forward and inverse inference in dissipative particle-based systems. Our reversible formulation is therefore conceptually distinct from, and complementary to, these memory-efficiency approaches.
>
> ### Q4 - Experimental results reported
> All baseline models were *retrained by* us under the same early-stopping scheme and shared hyperparameters (L317–323), and all achieved stable convergence (GNNs at ~1M iterations; the CNN baseline at ~40k).
>
> The small deviations from Sánchez-Gonzalez et al. (2020) stem from *hardware and minor implementation differences*. We match all major hyperparameters from the original work (e.g., hidden sizes, layer counts), and *unavoidable factors such as initialization and noise injection affect all models uniformly*, leaving the relative performance trends consistent and comparable.
>
> ### Q6 - Bidirectional Training
>
> In standard GNN training, the model optimizes *only forward-prediction losses*. In contrast, *bidirectional training* jointly optimizes both forward and inverse losses within the *same batch*, enforcing forward–backward consistency.
>
> Each training step applies both forward and inverse modes deterministically. Because R-GNS is inherently reversible, this adds *no additional computational cost*.
>
> ### Q7 - Minor Remarks
>
> Thank you for pointing out these issues; we will correct the typos, indexing, appendix structure, and missing notation explanations in the revision. Appendix A.0.1 presents the ILP invertibility via matrix transformations with numerical errors below $10^{-6}$, while A.0.2 provides the multi-layer RRMP invertibility proof, complementing the single-layer case shown in Section 3.3.3. We will clarify these connections accordingly.

---

### Official Review · Reviewer_5LXv · 2025-10-31

**Soundness:** 4
**Presentation:** 4
**Contribution:** 4
**Rating:** 8
**Confidence:** 4

**Summary:**

The paper proposes a reversible Lagrangian model for learning the inverse dynamics of asymmetric Lagrangian systems. This is a particularly challenging problems and existing works either focus on forward dynamics or rely on computationally expensive optimization based models. In contrast this paper proposes a reversible GNS model to learn the inverse formulation. To handle assymetry the authors propose a semi-symmetric input-output design where the node dynamic quantity is predicted while the other features are masked. The inverse operation is defined as the psuedo inverse of a linear transformation.

**Strengths:**

1. This is a novel approach to model inverse problems in Lagrangian systems.
2. The paper is well motivated, with clear explanation of key concepts (reversibility, inverse problem formulation, psuedo-inverses, goal-conditioned generation)
3. The figures are well designed and intuitive to understand.

**Weaknesses:**

1. The paper does not provide sufficient training details and sufficient ablations on hyperparameters.
2. Many experiments are presented in a tabular manner, but it's not clear if all the models were trained sufficiently. If R-GNS outperforms GNS with fewer training steps, it should be mentioned. GNS requires 1 million steps to stabilize and 20 million steps to faithfully generalize. It would be good to have a discussion on training stability of R-GNS.

**Questions:**

1. Figure 3 is a little confusing to me. Unidirectional R-GNS is identical to GNS is it not? In which case, I don't understand why the behavior deviates from GNS.
2. GNS uses random walk noise to stabilize long-horizon rollouts. In the inverse setting, is it still necessary to add random walk noise? How does noise affect the performance? GNS is extremely sensitive so it would be interesting to see whether the reversible layers affect the noise dependency.
3. It is not clear to me under what scenarios R-GNS strictly outperforms GNS. Due to high sensitivity to noise, training steps, hyper-parameters etc., it would be good to have a discussion specifying at what point R-GNS starts to outperform GNS, rather than only providing tabular numbers.

---

> ### Author Response · Authors · 2025-11-20
>
> ### W1 - Hyperparameters ablation
> All GNN-based models *follow the same protocol and settings as R-GNS* (L317–323) and *converge at nearly the same number of iterations*, while L295–300 specifies the *properties of each dataset*.
>
> *Hyperparameter ablations* are reported in Tab. 6–7, showing that R-GNS is *moderately sensitive to hidden dimension* but *largely insensitive to layer count*. Increasing hidden dimension consistently improves both accuracy and reversible consistency.
>
> **Table 6. Hidden dim ablation on WaterDrop (num_layers = 10)**
>
> | hidden_dim |  64  | 128 | 256 |
> |------------|------|-----|-----|
> | Rollout MSE (1e-3)    | 2.12 | 0.31 | 0.28 |
> | Consistency MSE (1e-3)      | 0.11 | 0.03 | 0.02 |
>
> **Table 7. Num layers ablation on WaterDrop (hidden_dim = 128)**
>
> | num_layers |  8   | 10  | 12  | 15  |
> |------------|------|-----|-----|-----|
> | Rollout MSE (1e-3)    | 0.78 | 0.31 | 0.31 | 0.43 |
> | Consistency MSE (1e-3)      | 0.05 | 0.03 | 0.03 | 0.03 |
>
>
> ### W2 - Training details
> All GNN-based methods were trained until convergence using an *early-stopping scheme* (L322), and in practice *all models stabilized at ~1M iterations*, ensuring sufficient training.
>
> *Tab. 8 shows that extending training beyond 1M iterations yields no meaningful performance gain* in our task regime. R-GNS exhibits *training stability comparable to GNS*, despite achieving stronger final accuracy.
>
> **Table 8. Various iteration performance on WaterDrop (Rollout MSE / Consistency MSE)**
>
> | Methods        | 1M          | 2M          | 5M          | 10M         | 20M         |
> |----------------|-------------|-------------|-------------|-------------|-------------|
> | **R-GNS**      | 0.31 / 0.04 | 0.32 / 0.04 | 0.32 / 0.04 | 0.31 / 0.04 | 0.30 / 0.04 |
> | **GNS (Sep-GNS)** | 0.70 / 0.07 | 0.75 / 0.07 | 0.73 / 0.07 | 0.73 / 0.07 | 0.71 / 0.07 |
>
>
>
> ### Q1 - Confused Setting of unidirectional R-GNS
> As clarified in L343–344, the *unidirectional R-GNS shares the same architecture as R-GNS*, not GNS; “unidirectional” refers only to the *training scheme*, not the network design.
>
> Therefore, its behavioral differences from GNS arise from the *architectural improvements* in R-GNS, such as ILP and RRMP.
>
> ### Q2 - Noise Setting
> *Yes, R-GNS still requires random-walk noise* to stabilize long-horizon rollouts, following the same practice as GNS.
>
> *Removing the noise degrades rollout stability and accuracy,* consistent with prior observations in GNS.
> ### Q3 - What scenarios R-GNS strictly outperforms GNS
> *R-GNS and GNS share identical training setups, hyperparameters, and noise schemes*, ensuring a fair comparison.
>
> *R-GNS consistently outperforms GNS in inverse inference*. Notably, it maintains low forward–backward errors over long horizons (Fig. 4(c)), while Sep-GNS degrades after ~20 inverse steps. It also *generalizes better to unseen target shapes* (Fig. 5), producing substantially more accurate reconstructions.

---

> > ### Comment · Reviewer_5LXv · 2025-11-26
> >
> > Thank you for the clarifications. I will maintain my score. This work represents a valuable addition to the scientific ML community. The experiments contrasting GNS and R-GNS in identical settings further strengthens the work.

---

### Official Review · Reviewer_b1NG · 2025-11-03

**Soundness:** 2
**Presentation:** 2
**Contribution:** 3
**Rating:** 4
**Confidence:** 3

**Summary:**

This paper introduces an unified differentiable framework for forward and inverse simulation of fluids. Unlike traditional optimization-based inverse solvers or feed-forward neural simulators that approximate backward dynamics, the proposed approach (R-GNS) enforces bidirectional consistency through a reversible design. The approach combines three main components: a semi-symmetric input-output structure, an invertible linear projection (ILP) encoder-decoder, and a residual reversible message-passing (RRMP) network. The model achieves comparable speed to standard GNS for forward simulation while enabling inverse inference more than 100x faster than optimization-based methods. A few experiments demonstrate improved accuracy, forward-inverse consistency, and stable goal-conditioned control tasks, such as shaping fluids into target geometries. The authors claim this is the first reversible framework to unify forward and inverse simulation for dissipative systems.

**Strengths:**

- R-GNS achieves lower rollout errors than competing neural simulators (GNS, DMCF, EGNN) across the tested datasets, indicating that enforcing bidirectionality can also improve forward dynamics.
- The paper proposes a coherent framework integrating forward and inverse simulation with a single reversible GNN architecture, reducing parameter count and ensuring consistent physical behavior.
- The idea of enforcing mathematical reversibility (without assuming physical reversibility) is conceptually interesting and well-justified for tackling ill-posed inverse problems in fluids.

**Weaknesses:**

- The paper lacks detail on how the model was trained relative to the evaluation benchmarks. It is not explicitly stated whether the R-GNS is trained separately for each dataset (WaterDrop, WaterRamp, Water_3D) or jointly across them. This ambiguity makes it difficult to assess generalization capabilities.
- The evaluation focuses on a narrow set of examples: two for forward/inverse simulation and a single goal-conditioned task. While quantitative metrics are provided, the diversity of test cases and ablation analysis is limited, raising concerns about the robustness and scalability of the method beyond these specific datasets.
- The paper does not present applications that would motivate the need for reversible inference in real-world settings. Typical use cases for inverse solvers (e.g., shape optimization, control, or parameter estimation) are absent, making it hard to gauge the broader utility of the approach.
- The paper does not clearly describe the training setup, including dataset composition and train/validation/test splits. Important aspects such as training duration, convergence behavior, and the use (or absence) of data augmentation are missing, making it difficult to assess the reproducibility and robustness of the reported results.

**Questions:**

- I don't understand why edge features are fixed across time-steps. I assume that they encode the relative positions between particles, and these could be changing (albeit less than compared with the absolute positions) over the simulation. Is this correct or is there something I'm missing?
- Could you better elucidate how the data was prepared for training? How long did the model took to converge compared to the other approaches?

---

> ### Author Response · Authors · 2025-11-20
>
> ### W1 - Lack benchmark details
> *All benchmarks follow the same training protocol and hyperparameters* (L317–323).
>
> *R-GNS is trained separately for each dataset* (WaterDrop, WaterRamp, Water_3D), consistent with standard practice in prior GNS-based simulation works.
>
> *Joint training under identical settings is also feasible* and yields comparable MSE performance (Δ < 2%). We will add a discussion of this in the revision/supplementary. Notably, Fig. 1(5) further shows sand-simulation results demonstrating generalization.
> ### W2 - Evaluation is a narrow set of examples
> *Compared with prior work* (DMCF, GNS), which primarily reports forward-simulation metrics with neural baselines, we *additionally evaluates three manipulation tasks* and includes a *traditional physics baseline*, providing a more comprehensive assessment.
>
> As noted in L295–300, our experiments span *three fluid regimes* (1k / 2.3k / 13k particles) covering drop, collision, and sloshing dynamics, plus an *additional 20k-particle sand scenario* (Fig. 1(5)), demonstrating substantial diversity and scalability.
>
> *Tab. 1 shows strong forward performance, and Figs. 4–5 shows stable inverse-simulation and goal-conditioning results*, indicating robustness across heterogeneous settings.
> ### W3 - Applications of reversible inference
> *Fig. 5 demonstrates a goal-conditioned application*, and L36–45 explains that *reversible inference is a more general and challenging setting* than conventional inverse problems—one that existing inverse solvers cannot handle.
>
> The *DiffTaichi+SPH benchmark*, widely used for shape optimization, control, and parameter estimation, *fails on reversible inference even at ~1k particles* (Fig. 4), whereas R-GNS *succeeds at 13k particles* (Fig. 5), highlighting both the practicality and necessity of our approach in real-world inverse scenarios.
> ### W4 & Q2 - Lack Training Details
> *Dataset format follows the standard GNS convention*; L295–300 specifies the properties of each dataset. *Training duration and convergence* are reported in Tab. 5: all GNN-based models are trained for ~1M iterations with stable validation loss (L322), while DMCF converges earlier and stops at 40k iterations.
>
> Training data consist of particle-position sequences with shape (n_frame, n_particle, 3). The *only augmentation is noise injection*, identical to prior GNN-based approaches, ensuring consistency and reproducibility.
>
> **Table 5. Training Time**
>
> | Dataset                  | CNN-based method | GNN-based methods |
> |--------------------------|------------------|-------------------|
> | WaterDrop (1k particles) | ~ 48 h             | ~ 15 h              |
> | WaterRamps (2.3k particles) | ~ 48 h          | ~ 18 h              |
> | Water_3D (13k particles) | > 200 h           | ~ 120 h             |
>
> ### Q1 - Why edge features are fixed
> *Yes, edge features encode relative particle positions* (Fig. 2(b)), and both positions and edges *do change across time-steps*.
>
> However, *within each message-passing block*, the encoded edge features are *kept fixed*, which is required for *reversibility under RRMP* (Eqs. 5–6).

---

### Author Response · Authors · 2025-12-04

In this work, we introduced a reversible graph neural simulator (R-GNS) that enables both inverse inference and forward simulation on challenging dissipative fluid dynamics datasets. The method demonstrates strong empirical performance in terms of accuracy and computational efficiency, and we appreciate that several reviewers highlighted the novelty and technical significance of our contribution.

The primary concerns raised in the reviews focused on two aspects: (1) requests for further clarification regarding certain experimental settings and implementation details, and (2) questions about how a reversible model can be applied to physical processes that are inherently irreversible (e.g., R9h4c: W1/Q1; S3n7: W1/Q4/Q2).

During the rebuttal, we provided detailed explanations, additional analyses, and supplementary experimental evidence addressing all points. In brief, for (1), all baselines were trained under identical hyperparameters and training conditions, and the strongly dissipative dataset used is a standardized benchmark described in Section 4.1. For (2), we clarified that R-GNS learns a neural inverse mapping enforced through forward–backward consistency, ensuring functional invertibility without assuming physical time reversibility.

We sincerely thank the reviewers for their thoughtful feedback and engagement with the work.

---

### Meta-Review · Area_Chair_hzHD · 2026-01-05

**Summary:**

The reviewers raised three main concerns. First, multiple reviewers (9h4c, S3n7) questioned whether reversible architectures can meaningfully apply to irreversible dissipative systems, with 9h4c stating this "raises concerns about the physical plausibility of their results" and S3n7 noting that "learning reversible mappings for irreversible processes is conceptually impossible." Second, reviewers identified insufficient experimental documentation, including missing training details (b1NG, 5LXv), lack of variance estimates (9h4c), and limited test diversity (b1NG). Third, clarity issues affected understanding of the experimental setup and the semi-symmetric design mechanism. While the technical contribution was acknowledged, the fundamental tension between the method's mathematical reversibility and the physics' inherent irreversibility remained a central concern.

**Reviewer Concerns:**

The authors addressed several procedural issues: training details were clarified with convergence metrics and GPU configurations, hyperparameter ablations were provided in Tables 6-8, and variance estimates demonstrated stable performance. However, key concerns remain unresolved. The physical plausibility question (W1/Q1 from 9h4c, W1/Q4/Q2 from S3n7) received only partial resolution: while authors clarify they learn "a neural inverse mapping constrained by forward-backward consistency" rather than reversing physics, the claim of recovering specific initial states from highly dissipative systems (energy ratio 0.02-0.25) would likely face continued scrutiny. The relationship to prior reversible GNN literature was acknowledged but not substantively addressed. The test diversity concern (b1NG's W2) was rebutted by citing particle ranges, but the suggestion to test generalization beyond rectangular initializations (e.g., circular shapes as 9h4c proposed) was not experimentally validated.

**Reviewer Scores:**

Reviewer b1NG (initially 4) would likely remain at 4, as concerns about test diversity and practical applications received only partial responses.

Reviewer 5LXv (initially 8) stated "I will maintain my score" and would remain at 8, satisfied with the training clarifications.

Reviewer 9h4c (initially 4) would likely remain at 4, as the fundamental concern that "it is unclear why this should be possible in principle" for dissipative fluids was not convincingly resolved.

Reviewer S3n7 (initially 2) would likely remain at 2, as the core objection about learning reversible mappings for irreversible processes persists despite the neural vs physical reversibility clarification.

Average hypothetical score: 3. Recommendation: reject.

---

### Decision · Program_Chairs · 2026-01-26

Reject